rsob.royalsocietypublishing.org

**Subject Area:**
cellular biology

transdifferentiation, microRNA, caspase, autophagy

**Author for correspondence:**
Tin Tin Su
e-mail: tin.su@colorado.edu

# Cellular plasticity, caspases and autophagy; that which does not kill us, well, makes us different

Tin Tin Su[1,2]

[1]Department of Molecular, Cellular and Developmental Biology, 347 UCB, University of Colorado, Boulder, CO 80309-0347, USA
[2]University of Colorado Comprehensive Cancer Center, Anschutz Medical Campus, 13001 E. 17th Pl., Aurora, CO 80045, USA

TTS, 0000-0003-0139-4390

The ability to regenerate is a fundamental requirement for tissue homeostasis. Regeneration draws on three sources of cells. First and best-studied are dedicated stem/progenitor cells. Second, existing cells may proliferate to compensate for the lost cells of the same type. Third, a different cell type may change fate to compensate for the lost cells. This review focuses on regeneration of the third type and will discuss the contributions by post-transcriptional mechanisms including the emerging evidence for cell-autonomous and non-lethal roles of cell death pathways.

## 1. Cellular plasticity in tissue homeostasis

Heterotopic ossification (HO) is a process in which bone grows ectopically during healing from heavily traumatized soft tissue such as damage common to wounded veterans. The description of HO dates back as far as the American Civil War and is extremely common in current conflicts, with prevalence of greater than 60% [1]. A search for the cellular origin of bone growth during HO has identified mesenchymal progenitor cells (MPCs) within traumatized muscle, which, after isolation, can differentiate into bone *in vitro* [2]. MPCs are not related to other muscle progenitor cells such as satellite cells, suggesting that they arose from muscle cells that underwent fate change. Thus, HO represents a profound example of cellular plasticity, which, in this case, can be quite detrimental to tissue repair.

Cellular plasticity in HO stands in contrast, to cell fates in adult organs that are typically stable, with any regeneration resulting from dedicated somatic stem cells (figure 1*a*). Because somatic stem cells can be identified readily, much of our knowledge about regeneration comes from experimental systems with dedicated stem cells: mouse hair follicle, planaria and *Drosophila* intestine, to name a few examples [3–5]. However, tissues without dedicated stem cells also regenerate. The vertebrate liver, for example, regenerates by proliferation of the surviving cells of each sub-type (figure 1*b*) [6–8]. A variation of this mechanism operates to regenerate the heart in zebrafish, wherein cardiomyocytes undergo limited de-differentiation, proliferate and re-differentiate into the same cell type [9]. If proliferation of hepatocytes is blocked during liver regeneration, however, biliary epithelial cells can de-differentiate, proliferate and re-differentiate into hepatocytes (figure 1*c*) [6–8]. Hepatocytes can do the same if proliferation of biliary epithelial cells is blocked. Such plasticity is observed also in other mammalian organs [10–12] and in some models of amphibian limb and fish fin regeneration [13].

A switch in cell identity and function or 'cellular plasticity' underlies regeneration regardless of the source of regenerative cells. The generation of different cell types by dedicated stem cells, *de-differentiation* from a differentiated state

rsob.royalsocietypublishing.org Open Biol. **8**: 180157

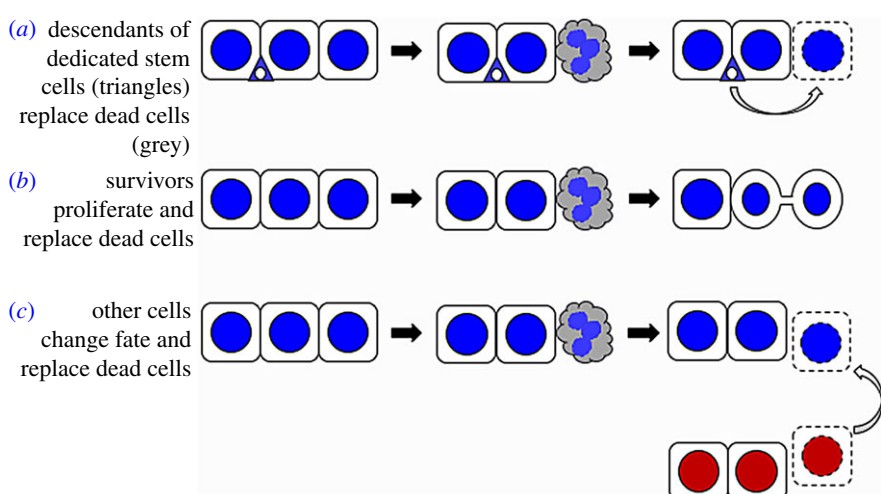

*(a)* descendants of dedicated stem cells (triangles) replace dead cells (grey)

*(b)* survivors proliferate and replace dead cells

*(c)* other cells change fate and replace dead cells

**Figure 1.** Three sources of regenerative cells.

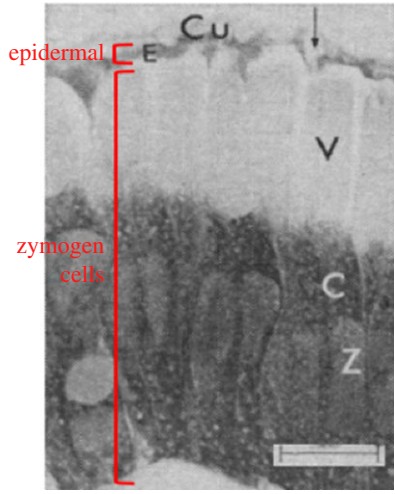

**Figure 2.** Transdifferentiation in silk moth. During metamorphosis, some of the squamous epithelial cells (E) that had been secreting the cuticle (Cu) during larval growth transdifferentiate into zymogen cells (Z) that secrete cocoonase. V, vacuole; C, cytoplasm. Arrow points to a duct. Modified from [15]. Scale bar = 20 μm.

back into the stem/progenitor state and *transdifferentiation* in which one differentiated state converts to another differentiated state are all examples of cellular plasticity. Because different definitions of each of these terms can be found in the literature, we will begin by defining the specific type of cellular plasticity to be discussed in this review, transdifferentiation.

## 2. What is transdifferentiation?

The term *transdifferentiation* was first coined by the eminent developmental biologist Fotis Kafatos in 1974 [14]. Kafatos had been studying the secretory cells of the silkmoths *Antheraea pernyi* and *Antheraea polyphemus*. During larval stages, epidermal cells in the silk gland are squamous in morphology and secrete proteins to make the larval cuticle. During metamorphosis, a subset of these cells retains their differentiated state to secrete the adult cuticle. But others transition, without dividing, into cells capable of secreting cocoonase enzymes that digest the cocoon and allow the pupa to escape. This transition is not simply a switch in gene expression from cuticle proteins to cocoonase. There are also marked changes in cell size, cell

shape and cell cycle regulation [15] (figure 2). Squamous epidermal cells become cuboidal in appearance, undergo endomitosis to increase ploidy and elongate as the cytoplasm fills with RNA-rich organelles. Even with these severe morphological changes, one could argue that these are simply outcomes of changes in gene expression. In other words, where do we draw the line between changes in gene expression in response to developmental needs and transdifferentiation?

A survey of the literature reveals many variations in the definition of transdifferentiation. Some define it as a process without an intermediate cell type, namely a direct conversion of one differentiated cell type to another (e.g. [16]). Others are agnostic and call it transdifferentiation even if there is an intermediate state involved (e.g. [17]). Some call it transdifferentiation only if initial and final cell types are within the same tissue, using the term 'metaplasia' when conversion occurs across tissues [18]. Others call it transdifferentiation even if it occurs across tissues [19]. Here, we will follow the more relaxed definition in which any conversion of one differentiated cell type to another differentiated cell type is transdifferentiation, as long as the experimental evidence meets the following three criteria. In their 2016 review, Merrell & Stanger [19] propose that for a transition to be called transdifferentiation: (i) transdifferentiating cells must be identified before and after the process, (ii) transdifferentiated cells must be functional at the new fate and (iii) transdifferentiated cells must be fully integrated at the tissue level. The secretory cells Kafatos described would fulfil all three criteria [14,15].

## 3. Molecular changes that accompany transdifferentation

Transdifferentiation that meets Merrell and Stanger's criteria occurs in at least the following three phyla, suggesting it is an evolutionarily conserved phenomenon: Nematoda (*Caenorhabditis elegans*), Arthropods (silkmoth, *Drosophila*) and Chordata (several members including frog, newt, zebrafish, mouse and human) (reviewed in [17,20,21]). Jellyfish (phylum Cnidaria) are described to also transdifferentiate because, in the case of stress, adult organisms can revert to an earlier developmental stage [22], but cell lineages have not been followed during this process *in situ* to ensure that fate changes satisfy the criteria

rsob.royalsocietypublishing.org Open Biol. **8**: 180157

described in the preceding section. Isolated striated muscle cells from jellyfish do transdifferentiate in culture into smooth muscle *in vitro* [23,24].

When considering molecular mechanisms that underlie cell fate changes, transcriptional regulation comes to mind first. This is perhaps because the first experimentally induced transdifferentiation was achieved by overexpressing a single transcription factor, MyoD, which converted fibroblasts into myoblasts [25]. Likewise, for converting fibroblasts into induced pluripotent stem cells in the laboratory, as few as three transcription factors are sufficient, for example, SOX2, NANOG and OCT4 [17,20]. These results, as well as our increasing appreciation of how epigenetic changes at the chromatin level accompany changes in cell fate, have led to the focus on transcriptional regulation at the DNA level as the primary driver of fate changes. It is clear that for a cell to adopt a new differentiated state, it must transcribe different genes. The question is whether physiological changes in transcription are *sufficient* for cellular plasticity. In other words, are said transcription factors at endogenous levels sufficient to induce fate change? While this question is hard to answer directly because endogenous levels of a protein can vary widely, one could phrase it differently to reach the answer. Are there instances where something else besides transcription/chromatin factors is *required* for fate change? If so, transcriptional regulation is clearly not sufficient in those instances. The literature suggests that the resounding answer to this question is YES.

## 4. Post-transcriptional regulators required for cell fate changes

MicroRNAs (miRNAs) have emerged as molecules that are neither transcription factors nor chromatin regulators, but are needed for cell fate changes. Many studies document the ability of miRNAs to enforce cell fate changes when ectopically expressed or overexpressed [26–28]. Fewer studies document their requirement in loss-of-function experiments. The best examples come from experiments addressing the role of miRNAs in normal development of model organisms, especially *C. elegans* [29]. *Caenorhabditis elegans* embryos progress through four larval periods, L1–L4, before moulting into adults. Each larval period is associated with stereotypical cell division patterns and differentiation events. We know that it is the same cells that switch from one programme of cell division/differentiation to another because of well-mapped cell behaviour in this organism such as apoptosis and cell lineage relationships. In *C. elegans* heterochronic mutants, typical patterns of cell division and differentiation for a given larval period remain unchanged but occur earlier or later [30]. In other words, cells in heterochronic mutants show temporal identities that are found in the same lineage but at other times in development. Two well-studied heterochronic genes, *lin-4* and *let-7*, encode miRNAs. *lin-4* enforces the switch from L1 to L2 [31]; *lin-4* mutants fail to terminate the L1 programme and instead repeat it numerous times. *let-7* acts later in development to enforce the L4-to-adult transition [32]. Likewise, cells in *let-7* mutants fail to switch to the adult programme and instead repeat the L4-specific programme [32]. Thus, lin-4 and let-7 represent clear examples where transcriptional changes are insufficient and post-transcriptional regulation must also contribute to cell fate changes.

Both *lin-4* and *let-7* are conserved in vertebrates (*lin-4* homologues are known as mir125). But it has been technically challenging to assess their loss-of-function phenotype because each is present in multiple copies throughout the genome. However, there is evidence for a collective requirement: knock-out of proteins needed to generate miRNAs, Dicer and Dgcr8, in mice produced embryonic stem cells (ESCs) that can self-renew but are defective in differentiation into different cell types [33,34]. In other words, like in *C. elegans*, post-transcriptional changes are required for fate changes in mammals.

Examples in *C. elegans* and mouse ESCs point to the requirement for miRNAs in cell fate changes during normal development and differentiation of stem cells, respectively. How about in transdifferentiation? miRNAs are induced or repressed during transdifferentiation in various models (e.g. [35,36]), and their overexpression can induce transdifferentiation, suggesting the potential of using miRNAs to reprogramme cells for therapeutic purposes [26–28]. But are miRNAs *required* for transdifferentiation? In a model of transdifferentiation of pre-B cells into functional macrophages by overexpression of the transcription factor C/EBPα [37], several miRNAs including miR34a and miR-223 are induced [38]. Inhibition of miR34a and miR-223 with antagomirs reduced transdifferentiation as detected by the expression of macrophage markers. Therefore, in this experimentally induced model, two miRNAs are partially required for transdifferentiation.

The effect of miRNAs on gene expression is inhibitory, by de-stabilizing the target mRNA and/or by reducing its translation. *lin-4* targets the mRNA for LIN-14 that is needed for L1-specific proliferative behaviour and cell fate; LIN-4 must be downregulated by *lin-4* for the cells to switch to the L2-specific programme [39]. Likewise, *let-7* targets the mRNA for LIN-41, which must be downregulated by *let-7* for cells to switch to the next programme [39]. In the model of pre-B lymphocyte-to-macrophage transdifferentiation [38], it is the lymphoid transcription factor Lef-1 that must be inhibited. The transcription factor used to force this transdifferentiation, C/EBPα, binds the promoter of Lef-1 to repress it. Lef-1 is also the target of miR34a and miR-223. Transcriptional repression of Lef-1 by C/EBPα is apparently insufficient for efficient transdifferentiation because miR34a and miR-223 are needed additionally to repress Lef-1 post-transcriptionally, as described in the preceding paragraph. One inference from these examples is that changes in the transcriptional profile are insufficient to establish a new fate in some instances. One must also inhibit mRNAs that are already made and associated with the old fate, which can explain the requirements for miRNAs (figure 3). Extending this logic, is it necessary to also inhibit proteins associated with the old fate? This may be where cell death pathways step in.

## 5. Cell death pathways play non-lethal roles in fate changes

Apoptosis requires caspases, a family of cysteine-dependent aspartate-directed proteases (for a recent review, see [40]). Caspases are made as inactive proenzymes that are activated by cleavage. In vertebrate cells, release of cytochrome *c* from the mitochondria in response to internal or external death stimuli results in the cleavage and activation of apical caspases. Active apical caspases cleave to activate effector caspases. Cleaved caspases are subject to an additional level of inhibitory

rsob.royalsocietypublishing.org   *Open Biol.* **8**: 180157

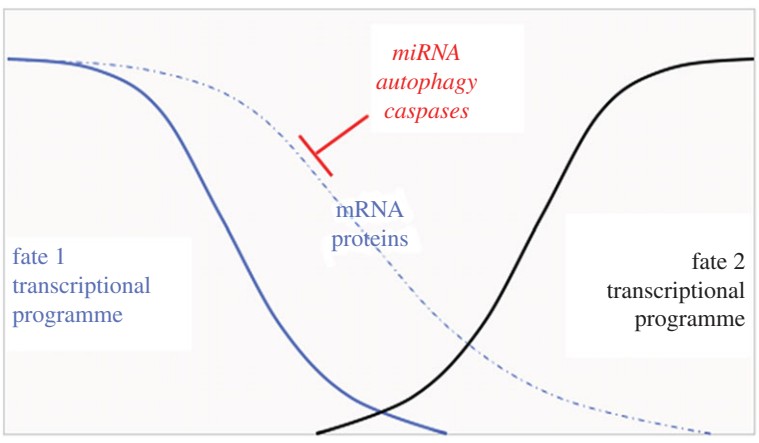

**Figure 3.** Transcriptional changes and post-transcriptional mechanisms enforce cell fate change. Changes in the transcriptional programme are only partially responsible because mRNAs and proteins associated with the old fate must also be erased. This task is accomplished by miRNAs (to neutralize mRNAs) and autophagy and caspases in non-lethal roles (to neutralize proteins).

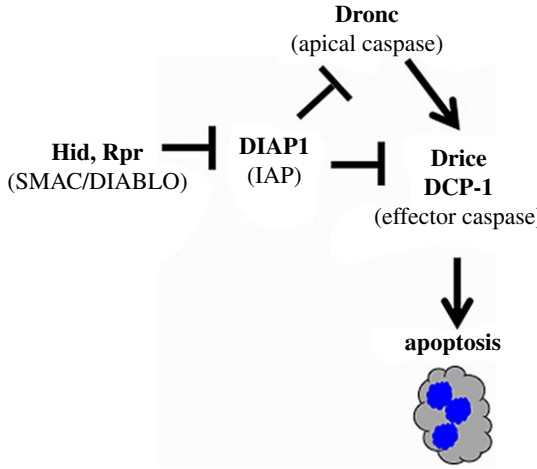

**Figure 4.** Apoptotic induction in *Drosophila*. Mammalian orthologues are in brackets.

regulation, by binding of inhibitor of apoptosis proteins (IAPs). Inhibition by IAPs is neutralized by SMAC/DIABLO proteins in vertebrates; SMAC stands for *second* mitochondrial activator of caspase, with cytochrome *c* being the *first*. Two key SMAC/DIABLO orthologues in *Drosophila* are encoded by *hid* and *rpr*. Ectopic induction of either protein is sufficient to induce apoptosis in *Drosophila* while the role of cytochrome *c* in apoptosis remains controversial in this organism [40] (figure 4).

Caspase activity is important also for non-apoptotic processes, as discussed in recent reviews [41–44]. Non-apoptotic roles for caspases fall into two buckets, cell non-autonomous and cell autonomous. An example in the first bucket is mitogenic signalling by apoptotic cells in a phenomenon known as apoptosis-induced proliferation (AiP) (reviewed in [45,46]). In *Drosophila* AiP, apical caspase Dronc cooperates with JNK signalling to produce secreted mitogenic signals that then promote proliferation of the surviving neighbouring cells. The role of Dronc in AiP occurs in addition to its role in activating effector caspases for apoptosis. In a variation of this process named 'phoenix rising', effector caspases 3 and 7 in mice cleave and activate calcium-independent phospholipase A2 to result in the generation and release of prostaglandin E, a known promoter of cell proliferation [47]. These are clearly cell non-autonomous effects; caspase activity in one cell changes the behaviour of another cell.

In the second bucket, caspases cleave proteins to affect cell behaviour in the same cell (reviewed in [41]). Examples include sperm tail individualization in *Drosophila* [48,49], cleavage and activation of Wg-inhibitor Sgg to temper Wg signalling during *Drosophila* neurogenesis [50], cleavage and activation of endonuclease G to result in genome instability and oncogenic transformation in human cells [51,52], and cleavage of NANOG to allow mouse ESCs to differentiate [53]. In the last study, mouse NANOG was cleaved by caspase 9 *in vitro*, cleavage products were detected in differentiating ESCs, and the expression of a caspase-resistant NANOG prevented ESCs from differentiating. Cell cycle inhibitor p21 is also a caspase substrate *in vitro* and in vertebrate cells, and cleaved p21 could not interact with one of its targets, replication factor PCNA, but the significance of p21 cleavage in cell behaviour remains to be investigated ([54], reviewed in [44]). We discussed previously the role of miRNAs as a post-transcriptional mechanism that enforces developmental switches in *C. elegans.* LIN-28 is another protein that must be downregulated for such a switch [39]. LIN-28 protein turns out to be a substrate for the sole *C. elegans* caspase CED-3 both *in vitro* and *in vivo* [55]. Worms expressing a caspase-insensitive LIN-28 show prolongation of some aspects of the L1/L2-specific programme, suggesting that caspase-dependent removal of LIN-28 contributes in part to terminating a developmental stage. Caspases are not alone in this task. First, CED-3 forms a complex and works together with Arg/N-end rule degradation pathway to remove LIN-28 [56]. Second, CED-3 was isolated as a genetic enhancer of developmental defects in miRNA mutants [55]. The emerging view is that miRNAs and caspase CED3, along with N-end degradation pathway, cooperate to erase old mRNAs and proteins as cells switch fate (figure 3).

The role of apoptotic caspases in cell fate plasticity is seen also during regeneration. Caspase 3 cleaves PAX7 in satellite cells (muscle stem cells) to terminate self-renewal and promote differentiation during muscle regeneration in mice [57]. During regeneration of amputated newt limbs, cells first de-differentiate to form a regenerative blastema. The blastema cells then proliferate and re-differentiate into appropriate cell types to form the new limb. During de-differentiation, myofibres that are multinucleate first fragment into mononucleated cells [58]. TUNEL stain, a marker of apoptosis, was detected during the initial stages of de-differentiation. Active caspase 3, by contrast, was detected throughout the 14+ days of

rsob.royalsocietypublishing.org  Open Biol. **8**: 180157

de-differentiation and in many myonuclei that lack the TUNEL signal, suggesting that cells activate caspases but do not die during de-differentiation. Blocking caspase activity in the myo-fibres by overexpressing an IAP blocked de-differentiation. Cells with active caspase 3 but no TUNEL staining are also observed during regeneration of zebrafish adult extraocular muscle, although the requirement for caspase activity has not been assessed in this model [59].

Is there evidence that caspase activity is required for transdifferentiation? Regeneration of *Drosophila* larval imaginal discs provides many examples of cell fate plasticity. Larval imaginal discs in *Drosophila* are made of a single-layer epithelium and lack a dedicated stem cell pool. *Transdetermination* in which cells of one imaginal disc switch into cells of another imaginal disc (e.g. leg-to-wing) during regeneration is considered a close parallel of transdifferentiation [60]. Recent studies using lineage tracing demonstrate fate conversion from one cell type to another within a single (wing) imaginal disc during regeneration. For example, when the wing pouch is ablated by the pouch-specific expression of a pro-apoptotic gene, nearby wing hinge cells translocate into the pouch, express pouch markers and help regenerate the pouch [61]. In our studies using X-ray doses that kill about half of the cells in the wing disc, the hinge was found to be protected from X-ray-induced apoptosis [62]. During regeneration, X-ray-resistant hinge cells lose hinge-specific gene expression, translocate into the pouch that suffers more X-ray-induced apoptosis, express pouch markers and participate in regeneration of the latter, much like in the genetic-ablation model. Inhibition of apical or effector caspases within the hinge during the X-ray-induced regeneration blocks both fate change and translocation, demonstrating a cell-autonomous requirement for caspase activity [63]. Because irradiated hinge cells do not die, the requirement for caspases may reflect a non-apoptotic role in cell fate plasticity. In support of this idea, we detect many cells that activate effector caspases without dying, as seen by a caspase-sensitive lineage tracer, in irradiated wing discs.

# 6. Autophagy in cell fate plasticity

Autophagy or 'self-eating' is an evolutionarily conserved process in which a cell encloses a part of its cytoplasm in a membrane-bound autophagosome. Autophagosomes deliver cellular parts to the lysosome for degradation (reviewed in [64,65]). Autophagy allows a cell to recycle parts in the case of nutritional stress and to remove portions of itself in the case of infection or damage. Genetic dissection of autophagy in yeast identified critical components including ATG (autophagy-related) proteins that assemble the autophagosome. ATGs are conserved from yeast to vertebrates. Knocking down ATG5 or ATG6 (also known as Becn1) interfered with regeneration in adult zebrafish [66]. In this model, after surgical removal of 50% of extraocular muscle, which controls eye movements, the remaining muscle regenerated to full anatomy and function. Lineage tracing experiments showed that residual myocytes were responsible for regeneration, as opposed to satellite cells (muscle stem cells) [66]. Myocytes are differentiated muscle cells with highly specialized cytoplasm filled with sarcomeres. Myocytes underwent de-differentiation into a mesenchymal state during regeneration, begging the question

of how the cytoplasm of myocytes is reprogrammed. The answer may be autophagy. Regenerating myocytes show double-membrane autophagosomes by electron microscopy and elevated expression of autophagy-related proteins such as ATG5. Depletion of ATG5 or ATG6 expression with morpholinos reduced the mass of regenerated muscle and caused disorganization of the regenerated cytoplasm [59].

Autophagy is required also for regeneration of the caudal fin in zebrafish [67]. In this model, surgical amputation of the tip of the tail is followed by de-differentiation of cells near the cut site to form a blastema. The blastema then re-differentiates to regenerate the fin. Cell types remain stable during this process, for example osteocytes de-differentiate to become part of the blastema, but re-differentiate into only osteocytes and not another cell type. Expression of ATG8-GFP (ATG8 is also known as LC3) increased in cells proximal to the cut and later in the blastema, as did the number of autophagosomes detected by electron microscopy. Depletion of ATG5 with a morpholino, as well as two drugs that are known to inhibit autophagy, prevented regeneration.

The above-described examples from zebrafish illustrate that autophagy is required to remodel the cytoplasm as cells change fate during regeneration. In both examples, the resulting cell type is the same as the originating cell type. Therefore, these are not examples of transdifferentiation. Changes in expression of autophagy-related genes have been detected in models of experimentally induced transdifferentiation (for example, [68–70]) and await functional studies.

# 7. Key remaining questions

The evidence for non-lethal roles of caspases leads to the question of how caspase activity may be restrained to allow cellular changes without killing the cell. The literature suggests mechanisms that regulate the level and sub-cellular localization of caspase activity. Genetic dissection of different effector caspases in *Drosophila* points to a threshold model in which the collective effector caspase activity must reach a threshold for a cell to undergo apoptosis [71]. In other words, cells below the threshold may have caspase activity but remain alive. A recent study in salivary glands suggests a mechanism for remaining below the threshold [72]. During *Drosophila* metamorphosis, salivary glands experience first a non-lethal dose of caspase activity, which cleaves cortical F-actin to alter cell morphology. This is followed by a later, higher dose that leads to cell death and dissolution of the gland. The two doses are controlled by hormone-induced waves of transcriptional factor activity that modulates the transcription of IAP antagonist Rpr, low for the first non-lethal dose and higher for the later lethal dose [72]. Related, in mouse ESCs, the level and speed of cytochrome *c* release from the mitochondria is suggested to define the level of caspase-3 activation [73]. Sub-cellular sequestration is another mechanism for generating live cells with caspase activity. In *Drosophila* AiP, Myo1D-dependent localization of apical caspase Dronc to the basal side of the plasma membrane allows for non-apoptotic signalling [74]. Likewise, in the example of salivary gland death described above, Rpr transcription leads to activation of Dronc but only at the cell cortex where cortical F-actin is remodelled.

Another key question is 'what are the non-apoptotic targets of caspases in fate change?' A few such as LIN-28, NANOG

and PAX7 are known, but there must be others and these likely differ between different cell types. Identifying and understanding their function would be required to understand how caspases enforce fate changes. This is especially true in the above-described models of regeneration in newt limb and *Drosophila* wing discs where caspase activity has been shown to be required. Identification of substrates in these models will distinguish between apoptotic and non-apoptotic contributions by caspases.

## 8. Relevance of cellular plasticity to human disease

In tumour biology, the concept of cancer stem cells has been controversial. But there is agreement that within a tumour, some cancer cells are better than others at re-initiating tumour growth, and are referred to as 'tumour initiating cells' or 'cancer stem-like cells' (CSCs) [75,76]. CSCs are also thought to be more resistant to treatment than cancer cells. Even if a treatment has successfully removed cancer cells, remaining CSCs could initiate a new tumour leading to recurrence. Therefore, eradication of CSCs is considered necessary for successful therapy, leading to efforts to identify agents that can effectively target CSCs. However, not only do CSCs generate non-stem cancer cells to initiate a new tumour, but non-stem cancer cells are now recognized as capable of converting to CSCs [78–81]. The plasticity that allows non-stem cancer cells and CSCs to interconvert presents a challenge to any therapy that targets CSCs. Another example of cellular plasticity that is relevant to tumour biology is epithelial–mesenchymal transition (EMT), which is considered a form of transdifferentiation that is highly relevant to tumour metastasis [77].

Interestingly, cancer treatments themselves promote the conversion of non-stem cancer cells into CSCs [78–81] and EMT-like behaviour [82,83]. Radiation and chemotherapy agents can activate caspases and induce autophagy (e.g. [84]), which could, in turn, promote cellular plasticity. Therefore, addressing the non-lethal roles of cell death pathways in cancer cell plasticity may identify new therapeutic targets to improve the treatment of cancer.

## 9. Conclusion

Gene expression status of a cell determines its identity. Transcription is the primary input into gene expression and changes in transcription underlie changes in cellular identity. But this is not the whole picture. Examples discussed here illustrate that post-transcriptional regulation of mRNAs, by miRNAs, and of proteins, by proteolytic activities normally associated with apoptosis and autophagy, also make essential contributions to switches in the identity of the cell. Cellular plasticity is required for normal development and for regeneration, and understanding it is important for understanding diseases such as cancer. It is through studies of both transcriptional and post-transcriptional mechanisms for cellular plasticity that we will fully understand the natural principles and devise better treatments for diseases.

Data accessibility. This article has no additional data.

Competing interests. I declare I have no competing interests.

Funding. T.T.S. is supported by a grant from the NIH (RO1 GM106317).

Acknowledgements. The author thanks Barbara Frederick, Nathan Gomes and Min Han for critical reading of the manuscript.

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
