## [Reviewer comments · Open Biology]

Review History

RSOB-18-0157.R0 (Original submission)

Review form: Reviewer 1

Recommendation

Major revision is needed (please make suggestions in comments)

Are each of the following suitable for general readers?

- a) **Title**
Yes
- b) **Summary**
Yes
- c) **Introduction**
No

Is the length of the paper justified?

Yes

Should the paper be seen by a specialist statistical reviewer?

No

Is it clear how to make all supporting data available?

Not Applicable

Is the supplementary material necessary; and if so is it adequate and clear?

Not Applicable

Do you have any ethical concerns with this paper?

No

Comments to the Author

This manuscript describes the author's thoughts on an interesting and timely topic, that of de-differentiation of cells for the purpose of tissue regeneration. The goal of the piece, stated in the abstract, is to discuss how cell fate changes contribute to tissue regeneration, with a specific focus on post-transcriptional mechanisms such as non-autonomous cell death, microRNAs, and apoptosis-independent caspase activity. These different mechanisms are introduced and their possible use in regeneration discussed in individual sections. Although the overall topic is interesting, I had difficulty following the logic of some sections and in other sections wished I was given more information.

The manuscript starts with a description of heterotypic ossification, or damage-stimulated ectopic bone formation, by way of introducing a regenerative process in which cells undergo a fate change to participate in tissue regrowth. My initial reaction was, what an odd and jarring way to introduce regenerative processes (references to the Civil War, wounded veterans, etc). Although I later (after reading the ms a few times) could see how HO relates to regeneration, it still seemed less appropriate here than as justification in a grant application for relevance to human health. If kept in this review, it would benefit from being more clearly integrated with the context of the rest of the manuscript.

The author raises interesting ideas – e.g., that different examples of cellular plasticity (de-differentiation, trans-differentiation, trans-determination, heterochronic cell fate changes etc.) are as important as dedicated stem cells for tissues to undergo regeneration. However the discussion of these different processes was often not easy to read. For example, a lot of space is given to the let-7- lin-4 family of heterochronic miRs. Is the idea that this type of regulation is similar to cell fate changes in regeneration? Or is it presented just to demonstrate how microRNAs function, and how they could also function in regeneration? It was not clear to me. The discussion on non-apoptotic roles for caspases could also use more information. I wished there was more information about the removal of cell cycle inhibitors and the cleavage of Nanog in mouse ES cells, both of which seem quite relevant. Likewise the section on autophagy, where interesting examples are cited but not explained in much detail. If this review is meant for scientists, more details seem appropriate. The final section (8) on relevance to human disease is quite confusing (e.g., lines 268-271), and needs clarification. Finally, an overall summary might help bring coherence to all of the ideas introduced in this review.

Decision letter (RSOB-18-0157.R0)

03-Oct-2018

Dear Dr Su

We are pleased to inform you that your manuscript RSOB-18-0157 entitled "Cellular plasticity, caspases, and autophagy; that which does not kill us, well, makes us different." has been accepted by the Editor for publication in Open Biology. The reviewer has recommended publication, but also suggest some minor revisions to your manuscript. Therefore, we invite you to respond to their comments and revise your manuscript.

Please submit the revised version of your manuscript within 14 days. If you do not think you will be able to meet this date please let us know immediately and we can extend this deadline for you.

- 1) A text file of the manuscript (doc, txt, rtf or tex), including the references, tables (including captions) and figure captions. Please remove any tracked changes from the text before submission. PDF files are not an accepted format for the "Main Document".
- 2) A separate electronic file of each figure (tiff, EPS or print-quality PDF preferred). The format should be produced directly from original creation package, or original software format. Please note that PowerPoint files are not accepted.
- 3) Electronic supplementary material: this should be contained in a separate file from the main text and meet our ESM criteria (see <http://royalsocietypublishing.org/instructions-authors#question5>). All supplementary materials accompanying an accepted article will be treated as in their final form. They will be published alongside the paper on the journal website and posted on the online figshare repository. Files on figshare will be made available approximately one week before the accompanying article so that the supplementary material can be attributed a unique DOI.

Online supplementary material will also carry the title and description provided during submission, so please ensure these are accurate and informative. Note that the Royal Society will not edit or typeset supplementary material and it will be hosted as provided. Please ensure that

the supplementary material includes the paper details (authors, title, journal name, article DOI). Your article DOI will be 10.1098/rsob.2016[last 4 digits of e.g. 10.1098/rsob.20160049].

4) A media summary: a short non-technical summary (up to 100 words) of the key findings/importance of your manuscript. Please try to write in simple English, avoid jargon, explain the importance of the topic, outline the main implications and describe why this topic is newsworthy.

Images

Data-Sharing

It is a condition of publication that data supporting your paper are made available. Data should be made available either in the electronic supplementary material or through an appropriate repository. Details of how to access data should be included in your paper. Please see <http://royalsocietypublishing.org/site/authors/policy.xhtml#question6> for more details.

Data accessibility section

Sincerely,

The Open Biology Team
<mailto:openbiology@royalsociety.org>

Reviewer's Comments to Author:

Referee: 1

Comments to the Author(s)

This manuscript describes the author's thoughts on an interesting and timely topic, that of differentiation of cells for the purpose of tissue regeneration. The goal of the piece, stated in the abstract, is to discuss how cell fate changes contribute to tissue regeneration, with a specific focus on post-transcriptional mechanisms such as non-autonomous cell death, microRNAs, and apoptosis-independent caspase activity. These different mechanisms are introduced and their possible use in regeneration discussed in individual sections. Although the overall topic is interesting, I had difficulty following the logic of some sections and in other sections wished I was given more information.

The manuscript starts with a description of heterotypic ossification, or damage-stimulated ectopic bone formation, by way of introducing a regenerative process in which cells undergo a fate

change to participate in tissue regrowth. My initial reaction was, what an odd and jarring way to introduce regenerative processes (references to the Civil War, wounded veterans, etc). Although I later (after reading the ms a few times) could see how HO relates to regeneration, it still seemed less appropriate here than as justification in a grant application for relevance to human health. If kept in this review, it would benefit from being more clearly integrated with the context of the rest of the manuscript.

The author raises interesting ideas – e.g., that different examples of cellular plasticity (de-differentiation, trans-differentiation, trans-determination, heterochronic cell fate changes etc.) are as important as dedicated stem cells for tissues to undergo regeneration. However the discussion of these different processes was often not easy to read. For example, a lot of space is given to the let-7- lin-4 family of heterochronic miRs. Is the idea that this type of regulation is similar to cell fate changes in regeneration? Or is it presented just to demonstrate how microRNAs function, and how they could also function in regeneration? It was not clear to me. The discussion on non-apoptotic roles for caspases could also use more information. I wished there was more information about the removal of cell cycle inhibitors and the cleavage of Nanog in mouse ES cells, both of which seem quite relevant. Likewise the section on autophagy, where interesting examples are cited but not explained in much detail. If this review is meant for scientists, more details seem appropriate. The final section (8) on relevance to human disease is quite confusing (e.g., lines 268-271), and needs clarification. Finally, an overall summary might help bring coherence to all of the ideas introduced in this review.

Author's Response to Decision Letter for (RSOB-18-0157.R0)

See Appendix A.

RSOB-18-0157.R1 (Revision)

Review form: Reviewer 1

Recommendation

Accept with minor revision (please list in comments)

Are each of the following suitable for general readers?

- a) **Title**
Yes
- b) **Summary**
Yes
- c) **Introduction**
Yes

Is the length of the paper justified?

Yes

Should the paper be seen by a specialist statistical reviewer?

No

Is it clear how to make all supporting data available?

Not Applicable

Is the supplementary material necessary; and if so is it adequate and clear?

Not Applicable

Do you have any ethical concerns with this paper?

No

Comments to the Author

I appreciate the additional work the author put into this review. There are still a few minor grammatical issues that I assume the copy editor will correct. Additional grammatical errors include: 'activity' needs to be changed to 'activation' in line 173; parentheses in line 281 don't seem appropriate; references are needed at the end of sentence in line 282;

Decision letter (RSOB-18-0157.R1)

29-Oct-2018

Dear Dr Su

We are pleased to inform you that your manuscript RSOB-18-0157.R1 entitled "Cellular plasticity, caspases, and autophagy; that which does not kill us, well, makes us different." has been accepted by the Editor for publication in Open Biology. The reviewer(s) have recommended publication, but also suggest some minor revisions to your manuscript. Therefore, we invite you to respond to the reviewer(s)' comments and revise your manuscript.

Please submit the revised version of your manuscript within 14 days. If you do not think you will be able to meet this date please let us know immediately and we can extend this deadline for you.

- 1) A text file of the manuscript (doc, txt, rtf or tex), including the references, tables (including captions) and figure captions. Please remove any tracked changes from the text before submission. PDF files are not an accepted format for the "Main Document".
- 2) A separate electronic file of each figure (tiff, EPS or print-quality PDF preferred). The format should be produced directly from original creation package, or original software format. Please note that PowerPoint files are not accepted.
- 3) Electronic supplementary material: this should be contained in a separate file from the main text and meet our ESM criteria (see <http://royalsocietypublishing.org/instructions-authors#question5>). All supplementary materials accompanying an accepted article will be treated as in their final form. They will be published alongside the paper on the journal website and posted on the online figshare repository. Files on figshare will be made available approximately one week before the accompanying article so that the supplementary material can be attributed a unique DOI.

Online supplementary material will also carry the title and description provided during submission, so please ensure these are accurate and informative. Note that the Royal Society will not edit or typeset supplementary material and it will be hosted as provided. Please ensure that the supplementary material includes the paper details (authors, title, journal name, article DOI). Your article DOI will be 10.1098/rsob.2016[*last 4 digits of e.g. 10.1098/rsob.20160049*].

- 4) A media summary: a short non-technical summary (up to 100 words) of the key findings/importance of your manuscript. Please try to write in simple English, avoid jargon, explain the importance of the topic, outline the main implications and describe why this topic is newsworthy.

Images

Data-Sharing

It is a condition of publication that data supporting your paper are made available. Data should be made available either in the electronic supplementary material or through an appropriate repository. Details of how to access data should be included in your paper. Please see <http://royalsocietypublishing.org/site/authors/policy.xhtml#question6> for more details.

Data accessibility section

Sincerely,

The Open Biology Team
mailto:openbiology@royalsociety.org

ditage Insights by clicking on the following link: <https://www.surveymonkey.com/r/author-perspectives-on-academic-publishing-royal-society>

This should take no more than 15 minutes and you will have the opportunity to enter a prize draw. We hope these results will provide us with valuable insights we can use to improve our service.

Reviewer(s)' Comments to Author:

Referee: 1

Comments to the Author(s)

I appreciate the additional work the author put into this review. There are still a few minor grammatical issues that I assume the copy editor will correct. Additional grammatical errors include: 'activity' needs to be changed to 'activation' in line 173; parentheses in line 281 don't seem appropriate; references are needed at the end of sentence in line 282;

Author's Response to Decision Letter for (RSOB-18-0157.R1)

See Appendix B.

Decision letter (RSOB-18-0157.R2)

30-Oct-2018

Dear Dr Su

We are pleased to inform you that your manuscript entitled "Cellular plasticity, caspases, and autophagy; that which does not kill us, well, makes us different." has been accepted by the Editor for publication in Open Biology.

Sincerely,
The Open Biology Team
mailto:openbiology@royalsociety.org

Appendix A

Dear Editors and reviewer,

Thank you for your time in assessing this manuscript. I really appreciate the encouraging comments and am thankful for the critical comments. I have revised the manuscript to address all comments as shown in detail below. I hope the revised manuscript is acceptable for publication.

Tin Tin Su

Reviewer's comments and author's responses

1. Although the overall topic is interesting, I had difficulty following the logic of some sections and in other sections wished I was given more information.

AU: I have made revisions, highlight in yellow, throughout the manuscript to provide clarification and additional information as requested.

2. Although I later (after reading the ms a few times) could see how HO relates to regeneration, it still seemed less appropriate here than as justification in a grant application for relevance to human health. If kept in this review, it would benefit from being more clearly integrated with the context of the rest of the manuscript.

AU: Please see changes in lines 31-34, made in order to better integrate HO to the rest of the manuscript.

3. The author raises interesting ideas – e.g., that different examples of cellular plasticity (de-differentiation, trans-differentiation, trans-determination, heterochronic cell fate changes etc.) are as important as dedicated stem cells for tissues to undergo regeneration. However the discussion of these different processes was often not easy to read. For example, a lot of space is given to the let-7- lin-4 family of heterochronic miRNAs. Is the idea that this type of regulation is similar to cell fate changes in regeneration? Or is it presented just to demonstrate how microRNAs function, and how they could also function in regeneration? It was not clear to me.

AU: Please see changes in lines 117-118, 125-125, and 127. Briefly, C. elegans miRNAs are discussed as examples of the importance of post-transcriptional regulation to cell fate change.

4. The discussion on non-apoptotic roles for caspases could also use more information. I wished there was more information about the removal of cell cycle inhibitors and the cleavage of Nanog in mouse ES cells, both of which seem quite relevant. Likewise the section on autophagy, where interesting examples are cited but not explained in much detail. If this review is meant for scientists, more details seem appropriate.

AU: Additional details on caspases and their targets have been added as lines 174-180. In the examples cited on autophagy in fate change, there was not really additional relevant details to provide beyond what was already there in the original manuscript, the detection of autophagy and the finding that inhibition of autophagy interfered with regeneration. Therefore, I have not made changes to the autophagy section.

5. The final section (8) on relevance to human disease is quite confusing (e.g., lines 268-271), and needs clarification.

AU: I hope that revised sentences in lines 276-280 clarified this part.

6. Finally, an overall summary might help bring coherence to all of the ideas introduced in this review.

AU: A conclusion paragraph has been added to the end.

Appendix B

Dear Editors and reviewer,

Thank you for the comments on the revised version. I am particularly indebted to the reviewer for a close reading of the manuscript. I have made the following changes in response to the reviewer's comments. I hope the manuscript is now suitable for publication.

Tin Tin Su

Reviewer's comments and author's responses

'activity' needs to be changed to 'activation' in line 173

Done.

parentheses in line 281 don't seem appropriate

Removed

references are needed at the end of sentence in line 282

References have been added (Ref. 78-81)